# Geologically younger ecosystems are more dependent on soil biodiversity for supporting function

Jiao Feng [1,2], Yu-Rong Liu [1,2,3] ✉, David Eldridge [4], Qiaoyun Huang [1,2], Wenfeng Tan [2,3] & Manuel Delgado-Baquerizo [5] ✉

Soil biodiversity contains the metabolic toolbox supporting organic matter decomposition and nutrient cycling in the soil. However, as soil develops over millions of years, the buildup of plant cover, soil carbon and microbial biomass may relax the dependence of soil functions on soil biodiversity. To test this hypothesis, we evaluate the within-site soil biodiversity and function relationships across 87 globally distributed ecosystems ranging in soil age from centuries to millennia. We found that within-site soil biodiversity and function relationship is negatively correlated with soil age, suggesting a stronger dependence of ecosystem functioning on soil biodiversity in geologically younger than older ecosystems. We further show that increases in plant cover, soil carbon and microbial biomass as ecosystems develop, particularly in wetter conditions, lessen the critical need of soil biodiversity to sustain function. Our work highlights the importance of soil biodiversity for supporting function in drier and geologically younger ecosystems with low microbial biomass.

Soil biodiversity is critical for the sustainability of multiple ecosystem functions such as nutrient cycling, organic matter decomposition, and plant production[1–3]. Thus, a growing number of studies have highlighted the pivotal role of soil biodiversity in supporting ecosystem functions from local to global scales and across ecosystems[3–5]. The mechanisms behind the biodiversity-ecosystem function (BEF) relationship are relatively intuitive: diverse soil biota can provide a broad array of functions that enable the complex depolymerization of organic matter, ultimately regulating the ingress of nutrients and energy into the soil ecosystem[3,4]. This may be particularly important in less productive (e.g., drylands) or early successional ecosystems wherein plant cover, soil organic carbon (C), microbial biomass and nutrient availability are low[6,7], and ecosystem maintenance depends on

the biodiversity-driven processes, including biological nitrogen (N) fixation, litter and organic matter decomposition and associated inputs of key soil resources[5,8,9]. However, the extent to which soil biodiversity supports critical ecosystem functions (i.e., soil BEF) might diminish with increasing ecosystem development, as a result of greater plant biomass production and larger pools of C and nutrients[7,10]. Moreover, changes in dominant soil taxa during ecosystem development[11,12], which are known to influence essential soil functions[7,12,13], could further suppress the positive soil BEF relationships. Thus, following millions of years of soil development (i.e., pedogenesis) or under conditions experienced in more productive ecosystems, soil function may become less dependent upon, and therefore decoupled from, soil biodiversity. Yet, how and why the

[1]National Key Laboratory of Agricultural Microbiology, Huazhong Agricultural University, Wuhan 430070, China. [2]College of Resources and Environment, Huazhong Agricultural University, Wuhan 430070, China. [3]State Environmental Protection Key Laboratory of Soil Health and Green Remediation and Hubei Key Laboratory of Soil Environment and Pollution Remediation, Huazhong Agricultural University, Wuhan 430070, China. [4]Centre for Ecosystem Science, School of Biological, Earth and Environmental Sciences, University of New South Wales, Sydney NSW 2052, Australia. [5]Laboratorio de Biodiversidad y Funcionamiento Ecosistémico. Instituto de Recursos Naturales y Agrobiología de Sevilla (IRNAS), CSIC, Av. Reina Mercedes 10, E-41012 Sevilla, Spain. ✉e-mail: yrliu@mail.hzau.edu.cn; m.delgado.baquerizo@csic.es

contribution of soil biodiversity to ecosystem function changes as ecosystem develops remains virtually unknown.

Herein, we hypothesize that the fundamental role of soil biodiversity to support ecosystem functions (soil BEF) will be less important in older and more productive ecosystems versus younger and less productive ecosystems. The reason is that as ecosystems get old, they naturally accumulate organic matter having an important legacy of nutrients and C that can be recycled within the ecosystem. However, in low productive and in young ecosystems lacking biologically fixed elements such as nitrogen and C, biodiversity is essential to ensure the entrance of resources into the soil system. Assessing the contribution of soil biodiversity to regulating multiple ecosystem functions (multifunctionality) as ecosystem develops across wide environmental gradients is critical to better incorporate knowledge of soil microbial processes into Earth system models, and to identify areas for soil biodiversity conservation under future environmental changes.

To address these knowledge gaps, we investigated the changes in soil BEF relationships across 87 globally distributed sites along 16 soil chronosequences, ranging in age from centuries to millennia (Fig. 1A; Supplementary Table 1). The datasets used in this study was retrieved from refs. 13–15. This data has been previously used to understand the changes in soil biodiversity[13], antibiotic resistance genes (ARGs)[15,16] and ecosystem properties[14] during ecosystem development. These soils come from different chronosequences with known soil ages[13], offering a unique opportunity to evaluate the influence of pedogenesis on soil BEF relationships. For each study site, five composite soil samples containing information on soil biodiversity and functions were available. These samples were used to estimate the local correlation coefficients (Spearman) between soil biodiversity and function (local soil BEF relationships determined within each ecosystem) (Methods). In particular, we investigated the relationships between the diversity (richness) of four typical soil organisms (invertebrates, protists, fungi and bacteria) and multiple ecosystem functions related to water regulation, organic matter decomposition, mutualism, nutrient cycling, plant pathogen control, and ARGs control (Methods). Overall, the within-site chronosequences encompass a wide range of origins (volcanic, sedimentary, dunes, and glaciers), climatic conditions (tropical, temperate, continental, polar, and arid) and vegetation types (forests, shrublands, grasslands, and croplands). Various environmental factors, including spatial, climatic, plant, edaphic, and microbial factors, may also influence the linkages between soil biodiversity and ecosystem functions[12,13]. Our study simultaneously considered all of these factors using structural equation modeling (SEM) and mixed-effects meta-regression models to test hypotheses on the mechanistic relationships for soil BEF as ecosystem develops.

## Results and discussion

Our work provides valuable evidence that soil biodiversity is more important for supporting function in younger and less productive ecosystems. However, we also found a negative correlation between soil BEF and substrate age indicating that older ecosystems are less dependent on biodiversity to support function, probably as a consequence of the organic matter and microbial biomass reservoir built over millions of years of soil development. Our results are important to understand the natural history of soil BEF relationships, and better forecast under what environmental conditions soil biodiversity is especially important when supporting function.

### Reduced within-site BEF relationships as soil develops

We first explored the distribution of our soil BEF data, and found that within-site soil BEF relationships generally follow a normal shape distribution according to Skewness, kurtosis, and Shapiro-Wilk analyses ($p > 0.05$; Supplementary Fig. 1; Supplementary Table 2). Results of linear mixed-effects model showed that within-site BEF relationship between multidiversity (the averaged richness of invertebrates, protists,

fungi and bacteria) and averaging ecosystem multifunctionality[6] is negatively correlated with soil age (Fig. 1B). This result accords with our hypothesis and suggests that the contribution of soil biodiversity to support functions wanes after millions of years of soil development. Similar reductions in soil BEF relationship were observed even when considering a range of independent multifunctionality indexes with multiple thresholds (multi-threshold multifunctionality, including >25%, >50%, >75% and >90% thresholds). Specifically, the steepest decline was observed at the threshold of 50% and 75% ($p < 0.01$), indicating that greater biodiversity tends to support a lower number of functions working at high levels. This multi-threshold approach effectively captures the number of functions while accounting for trade-offs and correlations among functions[17], providing robust evidence for the importance of soil biodiversity in sustaining fundamental functions in younger soils, such as soil respiration, decomposition, and nutrient cycling working at high level of function. This aligns with the Odum's ecological successional theory that species in the early stages of development exhibit broader niches. Consequently, in younger soils with lower soil biodiversity (Supplementary Fig. 2), the increase in species diversity contributes to more efficient resource utilization and facilitates the enhancement of multiple ecosystem functions. In well-established older soils, however, the contribution of soil biodiversity to support function may be less noticeable given the legacy of millions of years of organic matter and microbial biomass accumulation which can now feed the ecosystem with resources. Additionally, the higher soil biodiversity following organic matter accumulation may lead to functional redundancy of essential functions measured in this study, further contributing to the diminished soil BEF relationships in older soils[5].

Moreover, the weakened soil BEF relationship with increasing soil age holds true when considering multiple individual functions supported by soil biodiversity, including soil respiration, Arbuscular Mycorrhizal fungi (AMF) mutualism, and water holding capacity (WHC) (Fig. 1C). Collectively, these declines in within-site soil BEF relationships suggest that soil biodiversity plays a crucial role in maintaining measured fundamental functions in geographically younger ecosystems, regardless of multifunctionality operating at low or high levels. We acknowledge that, in our study, the choice of functions may influence the evaluation of soil BEF relationships. Therefore, we emphasize the necessity of incorporating variables targeting broader dimensions of ecosystem functions, such as biological N fixation, food production or policy (among many others), into the multifunctionality frameworks to reinforce the robustness of conclusions in this study. Nevertheless, this study provides an important case illustrating the relationship between soil biodiversity and multifunctionality during soil development by incorporating multiple fundamental ecosystem functions, including soil respiration, decomposition, nutrient cycling, water and climate regulation etc.

### Ecological context as a driver of within-site soil BEF relationships

We then employed combined analyses of SEM and mixed-effects meta-regression model to gain a system-level understanding on the influence of multiple environmental factors including climate, soil age, plant, edaphic and microbial factors in driving local soil BEF relationship (Fig. 2A; Supplementary Figs. 3 and 4). Of the multiple environmental variables assessed, our results revealed microbial biomass to be the most important driver, exhibiting negative correlations with soil BEF relationships, regardless of whether function was considered as an average or as independent multi-threshold multifunctionality (Fig. 2B). Increased plant cover and litter inputs over millions of years of soil development promote soil organic C and nutrient availability, thereby fueling the production of microbial biomass[7,14]. Earth system models have consistently highlighted the role of microbial biomass in decomposition and organic matter mineralization through the regulation of extracellular enzyme

production[18]. Consequently, as soil age increases, the necessity of soil biodiversity for supporting ecosystem function is speculated to diminish, indirectly through the promotion of plant cover, soil organic C, and ultimately microbial biomass (Fig. 2A; Supplementary Fig. 5). According to the Odum's theory on ecosystem succession, there is a shift from ecosystems where soil is constrained by the accumulation of soil C and nutrients to those with sufficient resources. We posit that this shift from the less productive (oligotrophic) to the more productive (eutrophic) establishes a resource buffer and releases the dependency of

fundamental ecosystem functions, such as decomposition and nutrient cycling, on soil biodiversity in the older and well-developed soils[19]. Support for this proposition comes from the negative associations between soil organic C and soil BEF relationships (Fig. 2B). Overall, our findings suggest that geographically younger soils exhibit a greater dependence on soil biodiversity to sustain ecosystem functions due to lower plant cover, soil organic C accumulation and microbial biomass production.

We further revealed that local soil BEF relationships are influenced by changes in climatic factors, with significant reductions in the

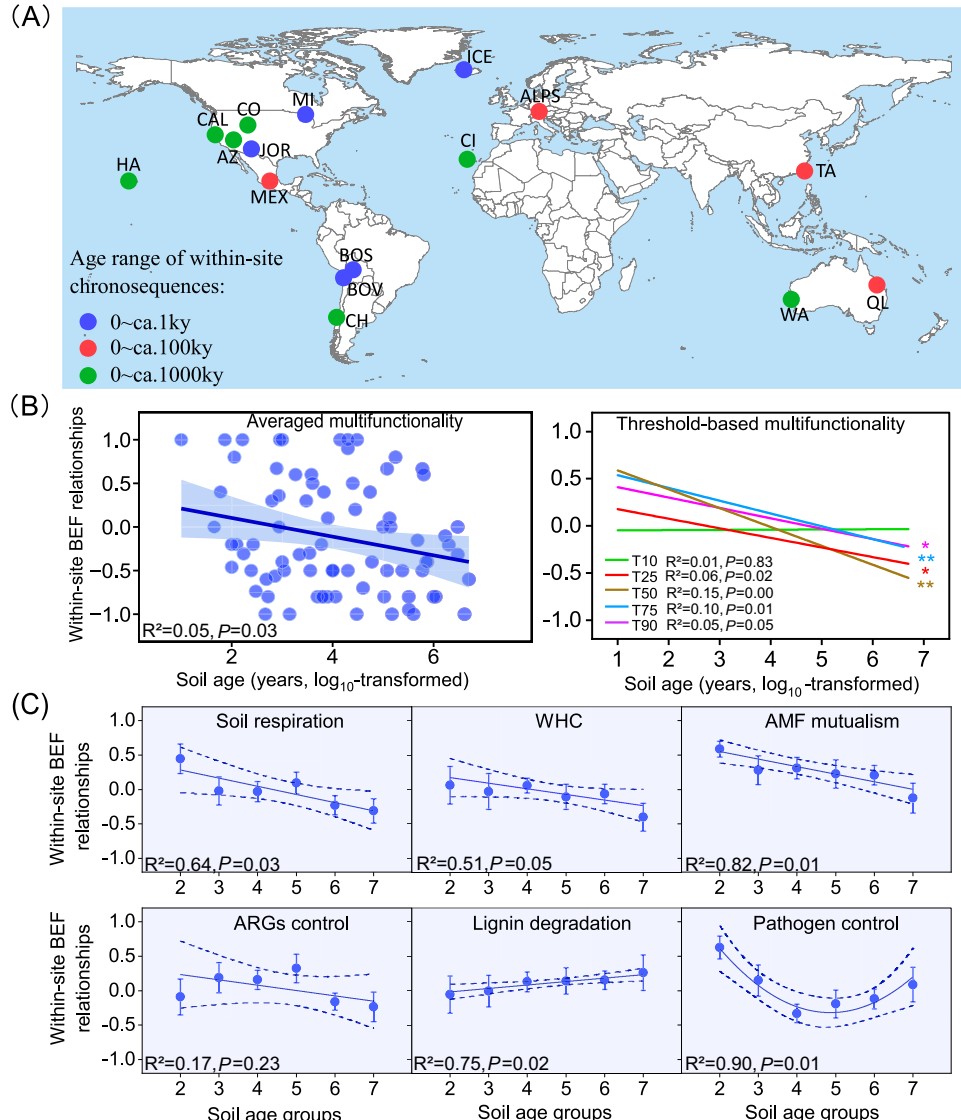

**Fig. 1 | Changes in the local (within-site) relationships between soil multidiversity and function (BEF) along different chronosequences. A** Locations of the 16 soil chronosequences (from 87 globally distributed sites) included in this study (see refs. 13–15 for original details); **B** Patterns of within-site BEF relationships between multidiversity and multifunctionality; and **C** Values of BEF between multidiversity and ecosystem individual functions for soils of different age groups. The free continental data of the world map in (**A**) was sourced from Natural Earth, supported by the North American Cartographic Information Society (https://www.naturalearthdata.com/). ArcGIS Desktop 10.8 (Esri, West Redlands, CA, USA) was employed for mapping the distribution of the study sites. Detailed information for the 16 chronosequences was shown in Supplementary Table 1. The values (mean ± standard error (SE)) of BEF relationships were calculated using the Spearman rank correlations (Methods). Multidiversity represents averaging biodiversity of invertebrates, protists, fungi and bacteria. Averaged, T10, T25, T50, T75 and T90, represents within-site BEF relationship

between multidiversity and multifunctionality quantified using averaged method and at threshold of 10%, 25%, 75% and 90%, respectively (Methods). The error bands surrounding the regression lines represent the 95% confidence interval of the correlation. The color of lines in (**B**) represents within-site soil BEF relationships between multidiversity and multifunctionality quantified using averaged and threshold approaches. In (**C**), soils were classified into 6 groups ranging of age from hundreds ($10^2$) to millions ($10^7$) of years in a power series: Group 2, < $10^2$ years, $n = 8$ independent samples; Group 3, $10^2$ - $10^3$ years, $n = 18$ independent samples; Group 4, $10^3$ - $10^4$ years, $n = 20$ independent samples; Group 5, $10^4$ - $10^5$ years, $n = 15$ independent samples; Group 6, $10^5$ - $10^6$ years, $n = 18$ independent samples; Group 7, $10^6$ - $10^7$ years, $n = 8$ independent samples. Ky, 1000 years; AMF Arbuscular Mycorrhizal fungi, ARGs antibiotic resistance genes. A two-sided test was used to assess the significance of the correlation analysis, with a threshold of $P$ value < 0.05 (*) and < 0.01 (**), respectively. Exact $P$ value and source data are provided as a Source Data file.

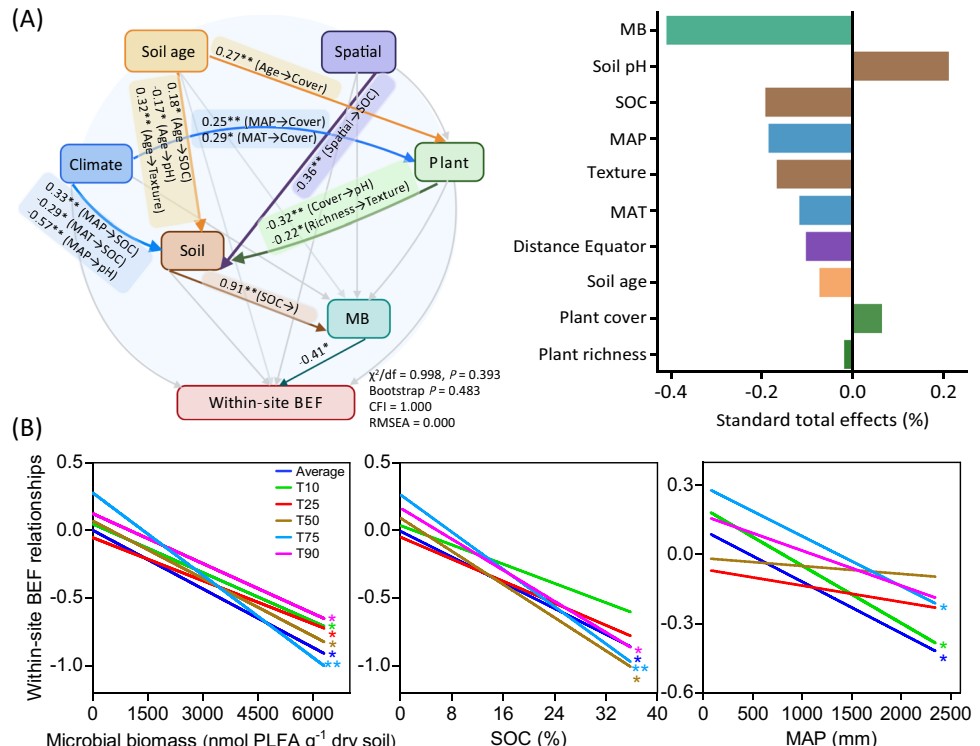

**Fig. 2 | Potential major factors influencing the local relationship between soil multidiversity and ecosystem function (BEF) along chronosequences.**
**A** Relative importance of soil age and different environmental factors (including microbial biomass, spatial, climatic, edaphic and plant attributes) regulating BEF relationships between multidiversity and multifunctionality using Structural equation model; and **B** Relationships of essential environmental factors with soil BEF relationships. Multidiversity represents averaging biodiversity of four groups of soil organisms, including invertebrates, protists, fungi and bacteria. Averaged, T10, T25, T50, T75 and T90, represents within-site BEF relationship between multidiversity and multifunctionality quantified using averaged method and at

threshold of 10%, 25%, 75% and 90%, respectively (Methods). The color of lines in (**B**) represents within-site soil BEF relationships between multidiversity and multifunctionality quantified using both averaged and threshold approaches. MB, microbial biomass, the sum of phospholipid fatty acid (PLFA) of bacteria and fungi. SOC, soil organic C; MAT, mean annual temperature; MAP, mean annual precipitation. Structural equation model was constructed using the maximum likelihood analysis to estimate the direct and mediating effect. A two-sided test was used to assess the significance of the correlation analysis, with a threshold of $P$ value < 0.05 (*) and < 0.01 (**), respectively. Exact $P$ value and source data are provided as a Source Data file.

relationship as mean annual precipitation (MAP) increases (Fig. 2B). By shaping the spatial patterns of plant cover and organic resources accumulation[5,20], MAP could indirectly influence the production of microbial biomass and thus the maintenance of ecosystem functions by soil biodiversity (Fig. 2A; Supplementary Fig. 5). Specifically, the decline in local soil BEF relationship with increasing MAP suggests that drier ecosystems are more dependent on the complementary utilization of resources by diverse taxa than more mesic regions. Consistently, our results show that local soil BEF relationships remain relatively constant across chronosequences in drylands, but declined significantly with soil development in non-drylands (Supplementary Fig. 6). Specifically, local soil BEF relationships remain constant or even became increasingly tight along local chronosequences such as in Jornada Desert and Cojiri. Conversely, more negative soil BEF relationships are observed at later stages of chronosequences in nondrylands, such as in Alps, Taiwan and Hawaii (Supplementary Fig. 7). In drylands, soils may be weakly developed even when aged due to water restrictions on soil weathering, soil organic C accumulation and biomass production[20,21]. Therefore, the negative influence of soil age on BEF may be limited in these ecosystems with reduced soil development. In other words, these ecosystems rarely accumulate enough organic matter and soil microbial biomass to be independent from soil biodiversity. Other factors influencing soil development, such as parent material may also exerting confounding effects on the patterns of within-site soil BEF along chronosequences. However, results of linear mixed-effects model showed that the observed declining pattern of within-site soil BEF with increasing soil age still held true after

accounting for the influence of the parent material (Fig. 1B). These results collectively underscore the pronounced reliance of multiple ecosystem functions on soil biodiversity in drylands, despite extensive soil maturation over millions of years. The revelation of this dependency amidst anticipated climatic shifts is especially pertinent, considering the increasing aridity of dryland ecosystems worldwide[21,22].

The BEF relationships between separate soil groups and multifunctionality with soil development is also likely linked to changes in plant cover, soil organic C and microbial biomass (Supplementary Figs. 8–12). For example, the BEF relationship between the richness of smaller prokaryotic bacteria and multifunctionality was negatively associated with microbial biomass (Supplementary Fig. 12), suggesting that the bacterial community may play a crucial role in sustaining multifunctionality in infertile, younger soils with low microbial biomass. This finding accords with previous studies highlighting the effectiveness of bacteria and bacterial-based energy channels in nutrient cycling and turnover, accounting for their relative dominance in the early stages of soil development[7,12]. Conversely, similar negative associations were absent when considering the local BEF relationships between larger eukaryotic soil groups, such as invertebrates and fungi, and multifunctionality (Supplementary Figs. 9 and 11). According to the Odum's theory, the body size of species increased during ecosystem development, accompanied by a gradual shift in life-history from an R-strategy (characterized by fast growth) to the K-strategy, which have higher competitiveness in a stable environment. Invertebrates and fungal communities that consume complex phenolic macromolecules and detritus (e.g., plant litter), have been found to be

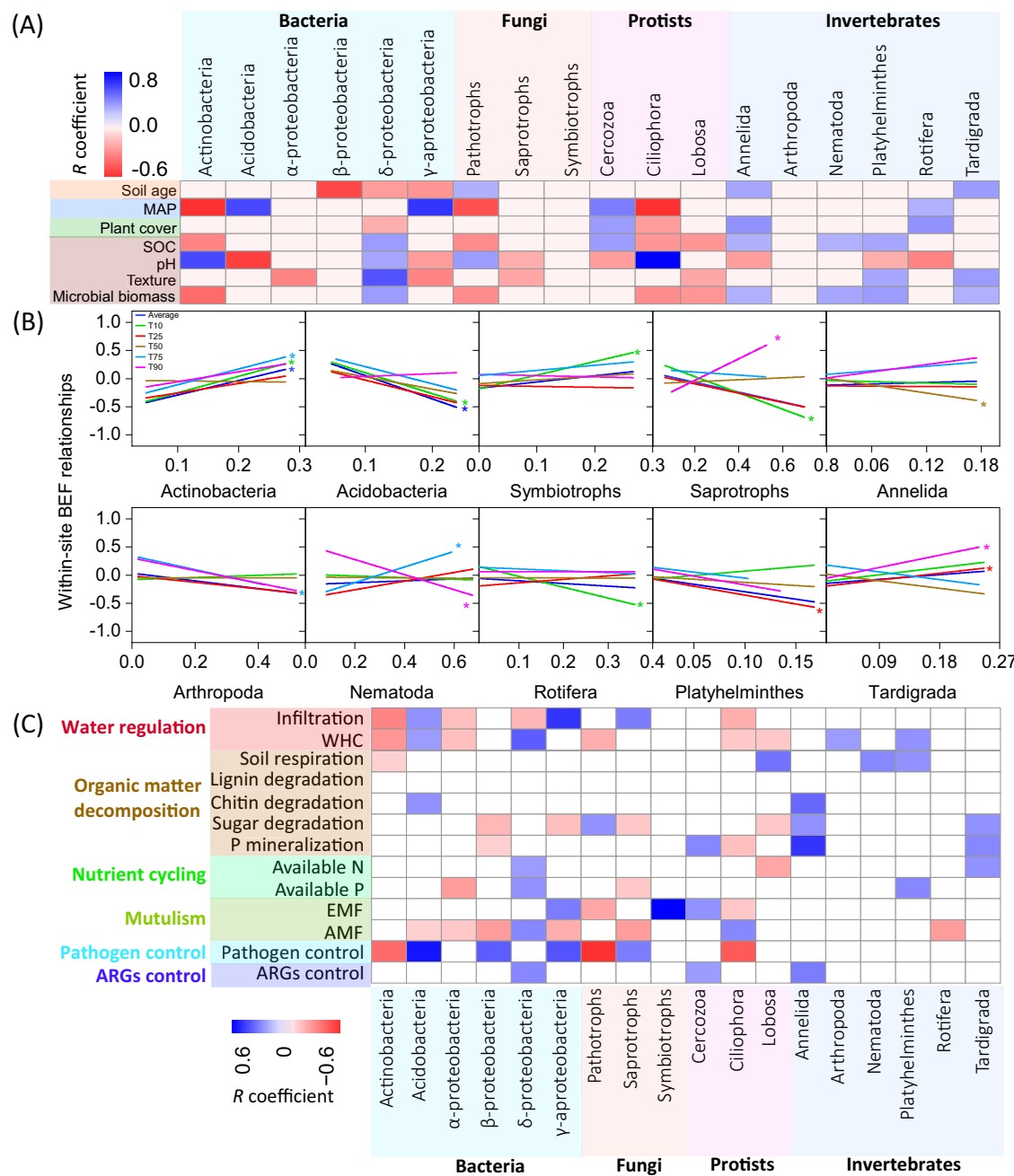

Figure 3

**Fig. 3 | The proportions of dominant soil organisms and their relationships with the local (within-site) soil multidiversity and ecosystem function (BEF) relationship along chronosequences. A** Relationships between the proportions of dominant bacterial, fungal, protistan and invertebrate taxa and essential environmental factors along chronosequences; **B** Correlations of the proportion of dominant soil organisms on BEF between multidiversity and multifunctionality; and **C** Relationships between the proportion of dominant soil organisms and multiple dimensions of ecosystem functions. Averaged, T10, T25,

T50, T75 and T90, represents within-site BEF relationship between multidiversity and multifunctionality quantified using averaged method and at threshold of 10%, 25%, 75% and 90%, respectively (Methods). The color of lines in (**B**) represents within-site soil BEF relationships between multidiversity and multifunctionality quantified using averaged and threshold methods. A two-sided test was used to assess the significance of the correlation analysis, with a threshold of P value < 0.05 (*) and < 0.01 (**), respectively. Exact P value and source data are provided as a Source Data file.

increasingly favored in long-term chronosequences[11,12], which could play essential roles in regulating the potential rates of multiple ecosystem processes. Accordingly, our study suggests a shift in the functional role of single soil groups in supporting ecosystem multifunctionality during long-term pedogenesis, highlighting the need for a multitrophic perspective when unravelling the drivers of soil BEF relationships under changing environmental conditions.

**Effects of soil community composition shifts on within-site soil BEF**

To gain further insights into the potential mechanisms underpinning changes in soil BEF relationships along soil chronosequences, we analyzed how shifts in the proportion of dominant taxa were associated with soil BEF relationships (Fig. 3). We found significant shifts in the community composition of soil organisms, including bacteria, fungi,

protists and invertebrates, across chronosequences and in response to changes in plant cover, soil organic C and microbial biomass (Fig. 3A). For instance, the proportion of larger soil invertebrates (e.g., Annelida, Nematoda, Platyhelminthes, and Tardigrada) increased significantly with increasing soil organic C and microbial biomass. These taxa are often winners under benign conditions[12,23], and may control the rates of multiple ecosystem processes by comminuting large amounts of plant residues and regulating resource flows within the brown food web[4]. Supporting this notion, we found positive correlations between the proportion of these dominant invertebrates and multiple ecosystem individual functions, including organic matter decomposition, soil respiration, phosphorus (P) mineralization, nutrient availability, and water regulation (Fig. 3C). Therefore, the promotion of these soil invertebrate taxa as soil ages may support multiple dimensions of individual functions, subsequently reducing the dependence of multifunctionality on biodiversity in older soils. In contrast, the proportion of Actinobacteria declined with increasing levels of soil organic C and microbial biomass, which showed positive correlations with local soil BEF relationships (Fig. 3B). In well-established soils with high microbial biomass, the dominance and performance of Actinobacteria in driving local soil BEF may therefore diminish. These findings align with previous studies suggesting that the abundance of Actinobacteria has significant positive effects on multifunctionality under infertile conditions[10,24]. Thus, a high level of overall biodiversity may not necessarily contribute equally to all measured functions in older and more complex ecosystems. Instead, certain functions may become more dependent on specific compositions within the community. These results underscore the pivotal role of soil community composition in driving fundamental ecosystem functions in older soils. In particular, we highlight the importance of conserving micro-faunal taxa for the maintenance of ecosystem functioning within these well-established soils.

## Implications of within-site soil BEF shifts along chronosequences

Our work provides important insights into the dynamics of within-site soil BEF relationships during long-term soil development, and stresses that climate, plant cover, soil C and microbial biomass play dominant roles in driving the long-term trajectories of ecosystem functions provided by multidiversity across biomes (Fig. 4). Changes in other factors such as soil pH, soil texture, and microbial network traits may also contribute to the reductions in soil BEF relationships as soil develops (Supplementary Figs. 13–15). For example, long-term pedogenesis frequently results in increases of finer soil particles and soil acidification[7,25]. These processes could act as environmental filters, leading to the selection of microbes and invertebrates with similar niches and, subsequently, niche overlap among soil taxa[12,13]. Furthermore, these factors can influence the relationships between the biodiversity of individual microbial groups and specific functions, resulting in both synergistic and trade-off effects on BEF between multidiversity and multifunctionality (Supplementary Figs. 8–12).

Additionally, it is essential to acknowledge that other undocumented factors operate at a local scale such as human activities (e.g., agriculture, deforestation), may also influence ecosystem succession and result in changes in soil BEF relationships. Importantly, disturbances of ecosystems can set back the development, causing a regression to early successional changes and leading to reductions in soil organic C and microbial biomass. Consequently, disturbances reducing soil organic matter and microbial biomass may rejuvenate the importance of soil biodiversity to support function. In our study, all locations were relatively undisturbed (with the exception of Taiwan) so future work is needed to delve further into this hypothesis. We further emphasize that for a more comprehensive understanding of the mechanisms driving soil BEF relationships during ecosystem development, it is crucial to simultaneously consider the successional trajectories of multiple factors, including disturbances in the local site and covariations in biotic and abiotic factors.

Collectively, our findings demonstrate the significant role of multidiversity in sustaining multifunctionality across various threshold demands and multiple individual functions in geographical younger and drier ecosystems. Additionally, we identified potential mechanisms underlying these findings. Specifically, plant cover, soil organic C, and microbial biomass, which increase as ecosystem age, appear to

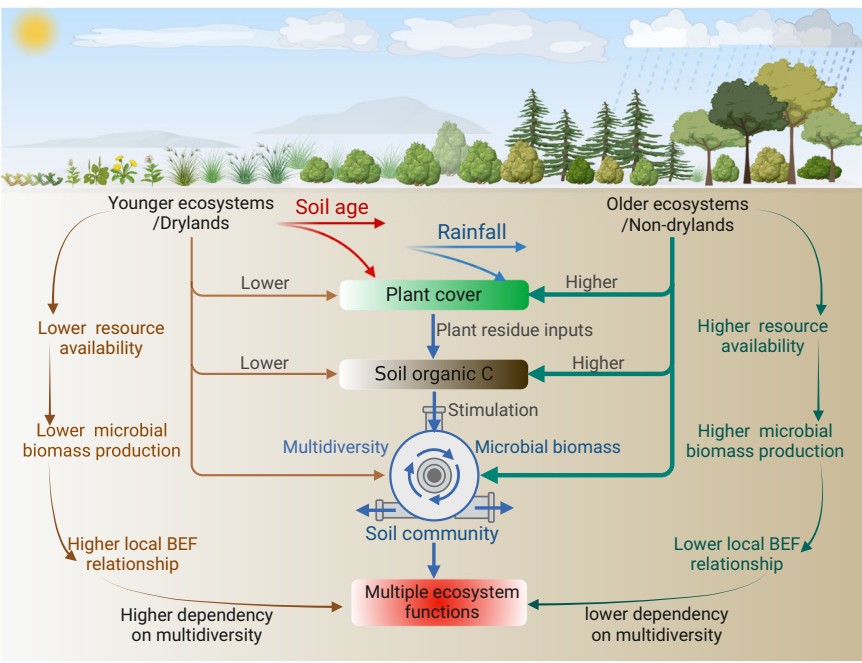

**Fig. 4 | A conceptual diagram illustrating the potential drivers of local (within-site) soil multidiversity and ecosystem function (BEF) relationship as soil develops across biomes in different climatic regions.** Multidiversity represents averaging biodiversity of four groups of soil organisms, including invertebrates, protists, fungi and bacteria. Figure 4 Created with Biorender.com released under a Creative Commons Attribution-NonCommercial-NoDerivs 4.0 International license (https://creativecommons.org/licenses/by-nc-nd/4.0/).

reduce the reliance of functions on soil biodiversity. Such findings are crucial as they enhance our understandings of how microbial processes regulate terrestrial ecosystem functioning under new climate regimes. Additionally, a deeper understanding of soil BEF relationships during soil development is relevant for improving model projections of Earth system models involved in biogeochemical cycles. Current models often lack explicit resolution of microbial processes over timescales, and only track the flows and mass balance of C and nutrients between different compartments of soil organic matter and microbial biomass[19,26]. Together, our work emphasizes the importance of multidiversity in sustaining ecosystem functioning, particularly in an increasingly drier and perturbed world.

## Methods

### Site information and sampling protocol

Data used in this study was mainly retrieved from refs. 13,14,15, all associated with the same survey. In these references, a global standardized field survey was conducted along 16 soil chronosequences between 2016 and 2017, which spanning nine countries from six continents (Fig. 1 and Supplementary Table 1). Soil ages along the selected chronosequences ranged from centuries to millennia according to geological surveys, isotopic dating techniques or models. In each chronosequence, soils differ only in the time since soil formation, with other soil-forming factors (including climate, vegetation, parent material and topography) remaining relatively constant (Supplementary Table 1). Overall, these chronosequences covered a wide range of chronosequence origins (volcanic, sedimentary, dunes, and glaciers), climatic conditions (tropical, temperate, continental, polar, and arid) and vegetation types (forests, shrublands, grasslands, and croplands).

Soil samplings were carried out following a standardized protocol as described in Ref. 14. In brief, a 50 m × 50 m plot was selected within each chronosequence stage, and then five composite surface (0-10 cm) soil samples were collected under the dominant vegetation types. Plot size was chosen to fully account for the spatial heterogeneity of different ecosystems (e.g., grasslands, shrublands and forests etc.). Plant cover and the number of perennial plant species (plant diversity) were surveyed and calculated. Following field surveys, soils were sieved in fields (< 2 mm) and separated into two subsamples. One soil subsample was air-dried to analyze soil physical and chemical properties. The other subsample of the soil was frozen immediately at −20 °C for molecular biology analysis. Taken together, a total of 435 soil samples from 87 plots, 16 chronosequences were analyzed.

### Methods of soil physical and biochemical analysis

Methodological information for these analyses is described in Ref. 14. In brief, for all soil samples, soil pH, soil salinity, soil texture (% of clay+silt), soil organic C, total N, total P and soil available P were measured. These soil variables were selected, because they have been found to change along chronosequences[7,14], and were associated with changes in soil community[25,27,28]. Soil pH was determined with a pH meter, in a 1:2.5 mass:volume soil and water suspension. Soil salinity was measured in an aqueous extract of saturated paste by a conductivity meter. Soil texture was analyzed using a hydrometer procedure. Soil organic C was determined by a colorimetric method after oxidation with potassium dichromate and sulfuric acid. Soil total N was analyzed using a CN analyzer (LECO CHN628 Series, LECO Corporation, St Joseph, MI, USA). Soil total P was measured using a digestion method with sulfuric acid (3h at 415 °C). Soil available P was determined using a colorimetric method after extraction with bicarbonate. The microbial biomass was expressed as the phospholipid fatty acids (PLFAs) extracted from freeze-dried soil samples[14].

### Methods of soil molecular analysis

Soil DNA extractions and sequencing was done as described in ref. 13. Soil DNA was extracted from soil samples using the Powersoil® DNA Isolation Kit (MoBio Laboratories, Carlsbad, CA, USA) following the manufacturer's instructions. To characterize the richness (number of phylotypes) of invertebrates, protists, fungi and bacteria, eukaryotic 18S rRNA and prokaryotic 16S rRNA genes were sequenced using Euk1391f/EukBr and 515F/806R pair sets[29,30]. Bioinformatic processing was performed using a combination of QIIME[31], USEARCH[32] and UNOISE3[33]. Phylotypes (i.e., Operational Taxonomic Units; OTUs) were identified at the 100% identity level. The OTU abundance tables were rarefied at 300 (invertebrates via 18S rRNA gene), 800 (protists via 18S rRNA gene), 2000 (fungi via 18S rRNA gene) and 5000 (bacteria via 16S rRNA gene), respectively, to ensure even sampling depth within each group of soil organisms. Protists were defined as all eukaryotic taxa, except for fungi, invertebrates (Metazoa), and vascular plants (Streptophyta).

### Soil biodiversity index calculation

The diversity (richness, i.e., number of phylotypes) of soil invertebrates, protists, fungi and bacteria was determined from rarefied OTU abundance tables. To obtain a quantitative index of soil biodiversity for each sample, the biodiversity traits of four groups of soil organisms (invertebrates, protists, fungi and bacteria) were combined by averaging the standardized scores (0-1 normalization) of diversity of all groups. This approach is commonly used to calculate multiple biodiversity indices (multidiversity) for soil and plant communities[6,34].

### Assessments of multiple ecosystem functions

This data was available from refs. 13,14 and 15. In each plot, 13 proxies reflecting ecosystem functions, processes, or properties regulated by soil organisms and belonging to a wide range of potential ecosystem functions were included: water regulation (potential infiltration, water holding capacity), organic matter decomposition (soil respiration and extracellular enzyme activities related to lignin, chitin, sugar degradation and P mineralization), nutrient cycling (available N, available P), mutualism (Ectomycorrhizal fungi (EMF) and Arbuscular Mycorrhizal fungi (AMF) fungi), plant pathogens control (reduced relative abundance of fungal plant pathogens in soil) and ARGs control (reduced abundance of ARGs in soils).

Data methods are described in refs. 13,14 and 15. In brief, the potential infiltration rate was measured by monitoring the time takes for a set amount of water to infiltrate through soil columns in the laboratory[35]. Soil water holding capacity was determined according to a saturation-drainage method[36]. The methods for available P were as described above. The availability of N (ammonium and nitrate) in soil was obtained by colorimetric assays after extracted using $K_2SO_4$ extracts. The activities of extracellular enzymes, including β-glucosidase (sugar degradation), N-acetylglucosaminidase (chitin degradation) and phosphatase (P mineralization) were measured using fluorescence method[37]. Moreover, a MicroResp approach was used to measure lignin-induced respiration[38]. Soil respiration (the basal flux of $CO_2$) was estimated using an isotope approach by adding $^{13}C$-glucose (99 atom% U-13C, Cambridge Isotope Laboratories)[39]. The relative abundances of EMF, AMF and potential fungal plant pathogens in soils were obtained from the amplicon sequencing analyses and were inferred by parsing the soil phylotypes using FUNguild[40]. ARGs were retrieved from ref. 15,16. The total abundance of 285 unique ARGs encoding resistance to all of the major categories of antibiotics was obtained by the high throughput qPCR[41]. The inverse abundance of potential fungal plant pathogens and ARGs were obtained by calculating the inverse of the variables, respectively (×−1).

To obtain a quantitative index for multiple ecosystem functions, multifunctionality was calculated by two of the most commonly methods used: "averaging approach" and the "multiple threshold approach"[17]. The averaging approach was evaluated by calculating the mean values of all 13 standardized (0-1 normalization) ecosystem functional proxies. Threshold-based approach evaluates the

total number of functions that exceed or equal to a predefined percentage of the maximum observed value of each individual function. Threshold-based multifunctionality (MF$_t$) was calculated using Eq. (1)[42]:

$$MF_t = \sum_{i=1}^{F} (r_i(f_i) > t_i) \qquad (1)$$

where F is the number of functions measured, $f_i$ represents the value for function i in a specific plot, $r_i$ represents a mathematical function that sets $f_i$ to be positive and $t_i$ is the threshold value corresponding to the predefined proportion of the maximum observed value for each function. Multiple threshold approach is commonly recommended, as the selection of a given threshold is arbitrary[17,34]. We used a set of threshold: 10%, 25%, 50%, 75% and 90% to represent multifunctionality that at low, medium, and high threshold of the observed maximum functioning.

### Microbial co-occurrence network traits

Microbial co-occurrence network including soil invertebrates, protists, fungi and bacteria was constructed for each plot, respectively, in order to assess the effects of pedogenesis on the overall architecture and potential biotic linkages of soil biota. A total of 87 co-occurrence networks were obtained to represent each chronosequence stages. The networks were constructed based on Spearman's correlation algorithm of the proportions of different phylotypes (OTUs). Specifically, networks were built and analyzed using the "WGCNA" and "igraph" R package. The nodes in the networks represent the taxonomic phylotypes of soil organisms, and the edges correspond to significant linkage between two nodes. A set of topological parameters were calculated to describe the biotic linkages of the networks, including the numbers of nodes and edges, positive and negative correlation numbers, average degree, graph density, degree centralization and modularity[43].

### Statistical analyses

The local soil BEF relationships were analyzed from multiple aspects: multidiversity vs. multifunctionality (both averaging approach and multiple threshold approach), biodiversity of separate groups of organisms (including invertebrates, protists, fungi and bacteria) vs. multifunctionality, and biodiversity of separate groups of organisms vs. individual ecosystem functions. In each plot, the spearman's correlation coefficients were calculated to represent the local soil BEF relationships[44,45]. Specifically, the correlation analyses were conducted using "corrplo" R package. The resulting P values were adjusted for multiple comparisons using the "fdr" method to control for the chance of false positives. Stronger positive correlation suggests higher capacity of soil biodiversity to promote ecosystem functions. The Skewness, kurtosis, and Shapiro-Wilk analyses were conducted to test the normality of local soil BEF relationships (Supplementary Fig. 1; Supplementary Table 2).

Linear mixed-effects model was employed to analyze the relationship of within-site soil BEF relationships between multidiversity and averaging ecosystem multifunctionality with soil age. In this model, the parent material, climatic conditions, vegetation type, and location were introduced as random factors using "nlme" R package. Moreover, soils were classified into 6 groups ranging of age from hundreds ($10^2$) to millions ($10^7$) of years, utilizing a power series based on soil ages provided in Supplementary Table 1. For instance, age group 2 represents soils with age lower than $10^2$ years, and age group 7 represents soils with age ranging between $10^6$ and $10^7$ years. Both linear and polynomial regression model were conducted to analyze the association between soil age groups and within-site soil BEF relationships. Furthermore, the association between soil age and BEF relationships were analyzed for drylands and non-dryland ecosystems,

respectively. Drylands and non-drylands were classified according to Aridity Index (AI): drylands (AI < 0.50) and non-drylands (AI > 0.5)[13].

We aimed to identify the best environmental variables as predictors of changes in BEF relationships during long-term pedogenesis. These environmental variables include multiple spatial (Distance equator), climatic (mean annual temperature, MAT; mean annual precipitation, MAP; mean diurnal range; precipitation seasonality, temperature seasonality), edaphic (soil texture, soil organic C, soil C:N ratio, soil pH, soil N:P, soil salinity), microbial biomass, plant (plant cover and plant richness) and microbial network traits (Node numbers, total edge numbers, positive edge numbers, negative edge numbers, average degree, degree centralization, graph density and modularity). Climatic data were obtained from WorldClim (http://www.worldclim.org) at a resolution of 1 km.

To achieve a system-level understanding of the potential major drivers for the local soil BEF relationships across spatial, time, edaphic and climatic gradients, a conceptual model was developed that could be further tested by SEM. The conceptual model was constructed according to previous knowledges (See Priori Model in Supplementary Fig. 3). We hypothesized that soil age, together with other important factors affecting pedogenesis such as climatic and biotic (including plant and microbial) attributes, may influence BEF relationships directly or indirectly by influencing edaphic variables and microbial biomass production[7,25,46]. The chi-squared test ($\chi^2$, the model has a good fit when $\chi^2$ was low and the P-value > 0.05) and the root-mean-square error of approximation (RSMEA, the model has a good fit when $0 \leq RMSEA \leq 0.05$) were conducted to test the overall goodness of SEM[47,48]. The SEM analysis was conducted by Amos 18.0 (IBM, SPSS, New York, USA).

Additionally, meta-analytic models were conducted to evaluate the combined effects of multiple environmental factors on within-site soil BEF relationships across chronosequences (Supplementary Fig. 4). Briefly, the mixed-effects meta-regression model was constructed by 'glmulti' package in R[49]. The importance of different factors was estimated according to the sum of Akaike weights. The weight was considered as the overall support for each variable in all potential models. A cutoff of 0.8 was set to identify the significant predictors for each model. Further, Pearson's correlation analysis was used to analyze the relationships of different environmental factors with local BEF relationships along chronosequences. A two-sided test was used to assess the significance of the correlation, with a threshold of P value < 0.05 and < 0.01, respectively.

### Reporting summary

Further information on research design is available in the Nature Portfolio Reporting Summary linked to this article.

## Data availability

The raw data and processes data for all figures and supplementary materials in this study have been deposited in the figshare under accession code: https://figshare.com/s/7999a7433ec52638a05b. Additional figures and tables can be found in the Supporting Information. Original data was retrieved from refs. 13,14 and 15.

## Code availability

All code associated with our analyses in this study is available at https://figshare.com/s/746673e2b49ba9374273.

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

## Acknowledgements
Wenfeng Tan was supported by the National Key Research and Development Program of China (2021YFD1901205). Jiao Feng was supported by the National Natural Science Foundation of China (32071595) and the Fundamental Research Funds for the Central Universities (2662023PY010). Yu-Rong Liu was supported by the National Natural Science Foundation of China (42177022). Manuel Delgado-Baquerizo acknowledges support from the Spanish Ministry of Science and Innovation (PID2020-115813RA-I00), and a project of the Fondo Europeo de Desarrollo Regional (FEDER) and the Consejería de Transformación Económica, Industria, Conocimiento y Universidades of the Junta de Andalucía (FEDER Andalucía 2014-2020 Objetivo temático "01 - Refuerzo de la investigación, el desarrollo tecnológico y la innovación") asociado with the research project P20_00879 (ANDABIOMA).

## Author contributions
M.D-B. and Y-R.L. developed the original ideas presented in the manuscript. J.F. and Y-R.L. analyzed the data. J.F. wrote the first draft of the paper. M.D-B., Y-R.L., D.E., Q.H. and W.T. edited the paper. All authors reviewed the paper and approved the final version of the manuscript.

## Competing interests
The authors declare no competing interests.
