## [Peer Review File · Nature Communications]

Geologically younger ecosystems are more dependent on soil biodiversity for supporting functionREVIEWER COMMENTS

Reviewer #1 (Remarks to the Author):

This manuscript presents a very interesting and novel study, testing the hypothesis that the relationships between soil biodiversity and soil functioning are stronger in younger soils than in older soils and in dryer soils than in wetter soils. Soil biodiversity – functions relationships were analysed across 87 ecosystems globally.

Results show that within site soil biodiversity-function relationships are negatively correlated with soil age thus suggesting a stronger dependence of soil functioning on soil biodiversity in geologically younger than older ecosystems. The same was found for dryer vs wetter ecosystems. Increases in plant cover, soil carbon and microbial biomass as ecosystems develop, and in wetter ecosystems, were found to reduce the importance of soil biodiversity-function relationships.

In my opinion, the study, and the results obtained, are worth publishing; the study is scientifically sound. The significance of the work lies in the relevance of understanding the relations between biodiversity and function in a context of climate-change and land use change, and possibly there are also implications for agricultural soils that are subject to intensive disturbance.

In general, the methods are described in sufficient detail and the conclusions are well supported by the data.

I do have a number of observations and concerns that I would like to see being addressed in a revised version of the manuscript. Those are listed below. I also share the annotated manuscript with more detailed comments, including some spelling corrections. I hope this will help authors to further improve their manuscript.

Ln 44, Ln 56, Ln 87, Ln 102, and throughout the remainder of the text. The authors use the terms “soil age” or “soil development”, “geologically younger” and “ecosystem development” interchangeably. However, soil age may not be the same as ecosystem development. Disturbances of ecosystems may set back the development, returning to early successional changes without resetting soil development. It would be good if the authors could make a clearer distinction between soil age and duration of ecosystem development as the two may not always coincide.

Ln 46 and Ln 116-118 and Ln 312-313: Could the authors please explain why Biological N fixation was not mentioned here, and not considered in the study.

Ln 92-94: In this context, the authors may also consider to briefly mention and refer to key literature on theory of ecosystems development / ecological succession and attributes related to biodiversity and function. See Odum EP (1969) The strategy of ecosystem development: An understanding of ecological succession provides a basis for resolving man’s conflict with nature. *Science* 164: 262–270. This would be

relevant to strengthen the Introduction and/or Discussion sections.

Ln 77, Ln 166-167 and section on statistical data analysis: Was the origin (or parent material) considered as a factor that could explain BEF relationships, in addition to age?

Ln 98-100 states “We found that within-site BEF relationship between multidiversity (richness of invertebrates, protists, fungi and bacteria) and averaging ecosystem multifunctionality is negatively correlated with soil age (Fig. 1B)”. Please note that for the average ecosystem functionality the R2 is extremely low. Do the data support this finding?

Ln 102: In the study described in this manuscript, only undisturbed ecosystems were studied? If not, does this sentence make sense?

Ln 126 and materials and methods – PLFA was analyzed on dried soil samples. What evidence do the authors have (and could be cited) do the authors have that soil drying before PLFA renders valid results?

Ln 176-177: The study included agricultural soils if I understood correctly, what do the results show about this relationship in agricultural soils (frequent disturbance, analogous to early successional stage, low Organic and microbial C)? In this sense, I would also encourage the authors to elaborate Ln 231-234 a bit more.

Ln 252-254: Please explain what original data the selection was based on.

Fig 1: The polynomial curve fitting doesn't seem to make any sense at least for the majority of those curves where a linear (or exponential) relationship seems to occur. Maybe only the figure on Pathogen control seems to be non-linear.

Ln 570: Would be good to know which functions related more or less strongly to biodiversity. Could the authors provide some results on individual functions rather than composite indices alone?

Reviewer #2 (Remarks to the Author):

Ref.: Ms. No. NCOMMS-23-49193

Geologically younger ecosystems are more dependent on soil biodiversity for supporting function
The study by Feng et al. examines the relationships between biodiversity and ecosystem function (BEF) in the soil across 87 ecosystems distributed globally, which encompass a wide range of soil ages, ranging from centuries to millennia. Their findings suggest that long-term soil development leads to a significant decoupling of the relationships between BEF. This implies that in younger ecosystems with less developed soils, there is a stronger reliance on soil biodiversity for proper soil functioning. Overall, this research provides valuable insights into the relationships between BEF as soil evolves over millennia

across a global scale. These insights have the potential to enhance our ability to predict how changes in soil biodiversity can affect various ecosystem functions in a dynamically changing world. The paper is generally well-organized and presents clear hypotheses. However, before its publication, the authors should clarify some areas of confusion in the results.

General comments:

1 The introduction section lacks critical information on the methodology of the study. For instance, there is a lack of clarity regarding the number of chronosequences included in this study and the specific methodologies employed to quantify the relationship between soil biodiversity and ecosystem functions.

2 The results regarding drylands are intriguing but require further clarification. The authors state that the capacity of soil biodiversity to support ecosystem functions will be weaker in older productive ecosystems, such as drylands (Line 57-59). However, it is essential to acknowledge that drylands do not uniformly consist of young soils with limited soil age. Drylands can have weakly developed or 'pedogenic young' soils due to weak leaching caused by water limitation. Therefore, even after millions of years of development, soils in drylands may remain weakly developed, potentially contributing to the non-significant reduction of soil BEF relationships across chronosequences in drylands. This explanation should be further elucidated throughout the manuscript.

3 The manuscript omits important definitions, such as 'multifunctionality', which should be provided at the first instance of their presentation in the text. It is imperative that these key terms are comprehensively defined and consistently used throughout the manuscript for the sake of clarity and precision.

Minor comments:

Line 40: Please define BEF for the first time of presence.

Line 50-51: Please define 'net soil functions'.

Line 52: As mentioned above, the term 'BEF relationship' should be defined for the first time of presence.

Line 59: While it is true that drylands are typically less productive, it is important to note that not all dryland soils are inherently young in terms of soil age. Drylands may exhibit 'young' pedogenic characteristics, primarily attributed to weak soil development as a result of poor leaching and water scarcity.

Line 61: Please define 'multifunctionality' for the first time of presence.

Line 65: Unclear. How many chronosequences do you have?

Line 71: Unclear. How the authors quantify soil BEF? Was Spearman's correlation coefficient used? Please clarify.

Line 86: This paragraph encapsulates the main findings of the manuscript. Therefore, it may be advisable to relocate the subtitle to the second paragraph.

Line 87: While drier ecosystems may have soils that are old in terms of their formation, they can also be geologically young due to weak soil development. It is necessary to provide further clarification on this matter.

Line 139: Please supplement the relevant references.

Line 165-167: This is an interesting point, but it requires further elucidation. The relatively constant soil BEF relationships along chronosequences in drylands may be attributed to the limited soil development in drylands.

Line 219: Other factors, such as soil pH, also varied significantly across chronosequences. Importantly, soil acidification as soil ages may also induce changes in soil microbial biomass and diversity. How will changes in soil pH influence soil BEF relationships?

Line 272: Please define the abbreviate for the first time of presence.

Line 369-370: The associated results should be provided in the Results section.

Line 376-377: Which models do you selected? Please clarify.

Line 394: Again, 'SEM' should be defined for the first time of presence.

Reviewer #3 (Remarks to the Author):

The manuscript by Feng et al. assess the effects of soil biodiversity on ecosystem functioning (water retention, decomposition, relative abundance of pathogens and mutualists) across 87 sites around the globe that vary in soil age (soil paedogenesis stages) which also co-vary with vegetation cover and biomass. The data show that geographically younger soils exhibit stronger positive soil biodiversity-ecosystem functioning relationships than older soils. The authors argue that this is largely because of the lower plant cover, soil organic C accumulation and microbial biomass production in younger dryland soils making them more susceptible reduced functioning with soil biodiversity loss. Conversely, in older soils there is more biological activity and inputs via plants that may relatively diminish the importance of soil biodiversity for ecosystem function. The study is very thorough in its data collection analyses and is generally well written. The study reveals important insights into the ecological function of soils, its biodiversity and community composition as ecosystems develop along a chrono sequence and in particular I think it can provide novel insights into the role of soil biodiversity and community composition with relation to fast versus slow ecosystem processes, and how ecological complexity increases with soil age. While I agree with statements emphasizing the need for soil conservation in drylands, there are a few points that need some clarification / consideration with regards to the diversity & redundancy vs compositional role in 'older' soils.

Main points

1. How comparable are the diversity gradients between younger and older soils? I assume older soils are more diverse? If biodiversity-function relationships are asymptotic then are the younger soils at the lower steeper end of the curve and older at the higher less steeper end of the curve?

2. On the issue of functional redundancy: This is an interesting point (L112-114), but I would be very cautious about how "redundancy" is used here for fear of miss-interpretation because redundancy fades with the consideration of temporal and spatial patterns in functioning and mode functions considered. Because there is no positive BEF relationship in some soils based on the specific functions measured does that mean that they are truly functionally redundant in the ecosystem, or that they the diversity and additional compositional components are not related to the functions you measured? The concern

here being that this may be interpreted to the general public as that soil biodiversity is not important in well established soils, thus we shouldn't be concerned about soil biodiversity in these systems. I don't think this is the message you are trying to convey. Theoretically, if there was a lot of true functional redundancy in that the organisms perform all the same functions in the same space at the same time would not resource competition lead to their divergent functional evolution? I think these statements in the text need to be changed for the relevance to the functions measured here with consideration for what "redundancy" means.

3. Alternative reasons for the lack of significantly positive BEF relationships in 'older' soils: L119-120: Ok. Or is it that in younger systems they are key for these functions, but in older ecosystems may play a greater diversity of functions not measured here and thus leading to a lack of strong BEF relationships based on the functions measured here. These sweeping statements can be misleading, when not all facets have been carefully considered. Then I see later on this is touched on L180-on that its not always biodiversity, but composition. It would be good to make this clear that diversity and composition may have varying roles as the soil ecosystem evolves where in resource poor biodiversity, having more, is better, while in older more complex ecosystems where there diversity is high composition is likely more important for given functions than just having more (i.e. the insurance hypothesis).

Minor points

L26-27: this sentence is awkward / incomplete. Rerword.

L41: I am not sure "tools" is really the appropriate term. Tools infers for use / manipulation. Why not "functions" or "biogeochemical processes"

L86: seems a bit abrupt to start with stating what the study shows without first stating the result. It would be better to start with L90 stating what the data show, then interpret for discussion point (eg. "our study could show that plant cover, soil C ...").

L91: Not clear to me what "increase in weather environments" means.. rerword.

L94-97: again this statement is meaningless here when there has been no evidence provided first to support these statements overselling the study without first presenting the facts.

L98: This is the important stuff to state first. The above paragraph adds nothing meaningful to the text and can be removed.

L105: This is inevitable based on the threshold method where the lowest threshold supports all functions across the diversity gradient and non are supported at all levels of diversity with the maximum threshold. What is interesting is the range in the thresholds and where it is at its maximum: Tmax, Tmin, and Tmode. Here it seems to be steepest at 50 and 75% indicating that greater biodiversity supports fewer functions above average as the soil ecosystem ages if I am interpreting this figure properly.

Fig. 1B. I am not sure you need a legend/specific colours for soil age when it is the x-axis. It could removed to declutter the panel.

L256 - Is soil age related to climate (Sorry if I missed it)? What if in Fig 2 climate predicted soil age (or vice versa or allowed to covary)? If climate is included as a co-variate (or data de-trended for climate) in the analyses change results?

L307: Not sure if these are all truly functions (i.e. enzyme activity) or ecosystem attributes (i.e. water holding capacity) that reflect its functioning. It's a picky thing that seems to come up these days, but simply stating that these measures 'reflect biological functioning' would help with clarity.

Responses to the reviewers

REVIEWER COMMENTS

Reviewer #1 (Remarks to the Author):

This manuscript presents a very interesting and novel study, testing the hypothesis that the relationships between soil biodiversity and soil functioning are stronger in younger soils than in older soils and in dryer soils than in wetter soils. Soil biodiversity – functions relationships were analysed across 87 ecosystems globally.

Results show that within site soil biodiversity-function relationships are negatively correlated with soil age thus suggesting a stronger dependence of soil functioning on soil biodiversity in geologically younger than older ecosystems. The same was found for dryer vs wetter ecosystems. Increases in plant cover, soil carbon and microbial biomass as ecosystems develop, and in wetter ecosystems, were found to reduce the importance of soil biodiversity-function relationships.

In my opinion, the study, and the results obtained, are worth publishing; the study is scientifically sound. The significance of the work lies in the relevance of understanding the relations between biodiversity and function in a context of climate-change and land use change, and possibly there are also implications for agricultural soils that are subject to intensive disturbance. In general, the methods are described in sufficient detail and the conclusions are well supported by the data.

I do have a number of observations and concerns that I would like to see being addressed in a revised version of the manuscript. Those are listed below. I also share the annotated manuscript with more detailed comments, including some spelling corrections. I hope this will help authors to further improve their manuscript.

1. Thank you for your positive and constructive comments. We aimed to address all your concerns regarding clarification and spelling errors you highlighted, and incorporated these important revisions to our manuscript. Your detailed comments provided in the annotated manuscript have also been addressed. Please see our revisions throughout the manuscript (e.g., lines 27-30, 46, 98-101, 127-134, 230-234, 312-313 etc.). Additionally, we have included texts to emphasize the implications of our conclusions in elucidating ecosystem development under disturbances in this revision (lines 252-261), and encouraging future work in this direction.

Ln27: change “develops” to “develop”.

2. Changed as suggested (line 27).

Ln28: Replace “contribution” by “importance”?

3. We have modified the sentence as suggested, by replacing “contribution” with “critical need” (line 28).

Ln 29-30: formulation could be simplified for the sake of readability. For example "in drier and geologically younger ecosystems, which are low in microbial biomass."

4. The sentence has been simplified as suggested (lines 28-30).

Ln 44, Ln 56, Ln 87, Ln 102, and throughout the remainder of the text. The authors use the terms "soil age" or "soil development", "geologically younger" and "ecosystem development" interchangeably. However, soil age may not be the same as ecosystem development. Disturbances of ecosystems may set back the development, returning to early successional changes without resetting soil development. It would be good if the authors could make a clearer distinction between soil age and duration of ecosystem development as the two may not always coincide.

5. Thank you for your insightful observations and critique. We acknowledge the importance of accurately distinguishing between "substrate age" and "ecosystem development" within our manuscript, as these concepts, while closely related, encapsulate distinct ecological phenomena.

We concur with your point that ecosystem development and soil development, despite their interconnectedness, may not always progress in parallel. Disturbances can indeed revert ecosystems to earlier successional stages without necessarily resetting the developmental trajectory of the soil. We have undertaken additional analyses using a mixed-effects model to better account for the confounding effects attributable to changes in vegetation (Fig. 1B). This allows us to more accurately assess the impact of disturbances on the within-site BEF relationships, which we have elaborated upon in lines 252-258.

To enhance the clarity of our discussion and results, we have made a conscious effort to consistently use the term "soil development" throughout the results section and in our subsequent discussion of these findings. This was done to ensure that we accurately delineate the outcomes of our research (e.g., lines 105, 133, 143, 178, 243, 269). Conversely, the term "ecosystem development" has been specifically reserved for use in the introduction section and when discussing the broader implications of our findings, thereby providing a comprehensive context for our study (e.g., lines 49, 50, 56, 259-260).

Ln 44: Geologically younger? Or early successional?

6. Revised it to "early successional" (line 43-44).

Ln 50: what are NET soil functions?

7. We have changed "net soil functions" into "essential soil functions" in the revised manuscript (line 51).

Ln 56: how to separate soil formation from ecosystem development (succession)?

8. As responded above, we acknowledge that soil formation and ecosystem development might occur asynchronously. Additional mixed effect model analyses were performed to address this point and to scrutinize the impact of disturbances on within-site soil biodiversity-ecosystem function (BEF) relationships (Fig. 1B, lines 252-258). Moreover, the term "soil development" is consistently utilized in describe and elucidate the main results, in order to precisely articulate the observations and outcomes of our study (refer to lines 105, 133, 143, 178, 243, 269). In comparison, "ecosystem development" is used exclusively in the introductory section and in the discussion of our work's broader implications. This distinction allows for a comprehensive and contextualized understanding of our research (see lines 49, 50, 56, 259-260).

L57: capacity is the right word here?

9. We have changed "capacity" into "contribution" (line 57).

Ln 58: add comma between older and productive; add comma between younger and less productive.

10. We have revised the sentence as "will be less important in older and more productive ecosystems versus younger and less productive ecosystems." (line 58-59).

Ln 65-66: ecosystem or soil age?

11. We have revised the sentence into "we investigated the changes in soil BEF relationships across 87 globally distributed sites along 16 soil chronosequences, ranging in age from centuries to millennia" to avoid any potential confusion between "ecosystem development" and "soil age" (lines 68-70).

Ln83: Or 'to test hypotheses on mechanistic relationships?'

12. Revised as suggested (lines 86-87).

Ln87: synonymus with soil age?

13. As responded above. Ecosystem development may not always coincide with soil development. We consistently used the term "soil development" in the results section and in our subsequent discussion of these results to precisely describe the findings of our study.

Ln 46 and Ln 116-118 and Ln 312-313: Could the authors please explain why Biological N fixation was not mentioned here, and not considered in the study.

14. Thanks for this insightful comment. We fully recognize the critical role of biological N fixation in the assessment of ecosystem functionality. However, obtaining quantitative estimates of biological N fixation on a large scale, particularly at the global scale, remains challenging. This challenge could be attributed to that virtually all approaches used for the measurement of biological

N fixation, including acetylene reduction assay, $^{15}\text{N}_2$ feeding or ^{15}N natural abundance methodology, are susceptible to specific potential sources of error when being applied to either symbiotic or non-symbiotic systems ¹⁻². Consequently, relying solely on one method for assessing biological N fixation at a global level, which encompasses both symbiotic and non-symbiotic N_2 -fixing systems, is not recommended. This may result in the lack of reliable comparisons of biological N fixation across different global ecosystems, leading to its exclusion in this study.

Additionally, the primary objective of this study was to investigate variations in the relationship between soil diversity and multifunctionality as soil develops. For this purpose, we included an array of fundamental ecosystem functions such as soil respiration, decomposition, nutrient cycling, and climate regulation. We acknowledge that the chosen functions could potentially impact the assessment of the relationship between soil biodiversity and multifunctionality. While incorporating these fundamental functions, we concur that broader dimensions of ecosystem functions, including biological N fixation, need to be integrated into the multifunctionality frameworks to provide a more holistic understanding. Despite our focus on these selected functions, this study serves as a foundational stepping-stone, demonstrating the possibility of multifunctionality assessments through the integration of various key ecosystem processes, such as soil respiration, decomposition, nutrient cycling, and climate regulation, amongst others. We have carefully discussed these limitations in the revised manuscript (lines 114-120).

Ln 92-94: should be linked to ecosystem theory. In this context, the authors may also consider to briefly mention and refer to key literature on theory of ecosystems development / ecological succession and attributes related to biodiversity and function. See Odum EP (1969) The strategy of ecosystem development: An understanding of ecological succession provides a basis for resolving man's conflict with nature. *Science* 164: 262–270. This would be relevant to strengthen the Introduction and/or Discussion sections.

15. We appreciate the valuable comments and suggestion to connect our work with the theory of ecosystems development and ecological succession. We have incorporated Odum's theory into both the Introduction and Discussion section. Changes can be found in lines 59-61, 114-120, 149-155, and 196-199. For instance, we stated that "According to the Odum's theory on ecosystem succession, there is a shift from ecosystems where soil is constrained by the accumulation of soil C and nutrients to those with sufficient resources. We posit that this shift from the less productive (oligotrophic) to the more productive (eutrophic) establishes a resource buffer and releases the dependency of fundamental ecosystem functions, such as decomposition and nutrient cycling, from soil biodiversity in the older and well-developed soils. (lines 149-155)."

Ln 77, Ln 166-167 and section on statistical data analysis: Was the origin (or parent material) considered as a factor that could explain BEF relationships, in addition to age?

16. We understand your concern. We have re-analyzed the data utilizing a mixed-effect model implemented through the “nlme” package. In this model, the parent material was introduced as a random factor. Our re-analysis confirms that the previously observed declining patterns of soil BEF with increasing soil ages remain consistent, even when accounting for the influence of parent material (Fig. 1B). These findings suggest that the parent material does not significantly alter the general trend of the BEF relationship across soil chronosequences. We have discussed the method of mixed-effect model and influences of parent material in lines 177-182, 400-402.

Ln 98-100 states “We found that within-site BEF relationship between multidiversity (richness of invertebrates, protists, fungi and bacteria) and averaging ecosystem multifunctionality is negatively correlated with soil age (Fig. 1B)”. Please note that for the average ecosystem functionality the R² is extremely low. Do the data support this finding?

17. We appreciate your concern. The low R² observed for average ecosystem functionality may be reflective of the intricate interactions within ecosystem. Indeed, ecosystems inherently exhibit complexity, influenced by a myriad of biotic and abiotic factors along large-scale chronosequences across the globe. We tried to deal with this challenge by employing a mixed-effect model. Specifically, we incorporated multiple factors—namely, parent material, climatic conditions, vegetation type, and location—into the mixed-effect model as random factors (lines 100-101, 177-182, 400-402). The inclusion of these factors marginally improved the R² of the relationship between soil BEF and age (Fig. 1B). This improvement indicates that considering a broader range of factors can enhance our understanding of the dynamics of soil BEF relationships during pedogenesis.

Nonetheless, it is clear that additional research is necessary to identify and incorporate other underlying factors that contribute to the variability observed. Exploring these avenues could help to enhance the robustness our findings and more accurately capture the complexity of ecosystem interactions. Despite the low R², the significant negative correlation is noteworthy ($p < 0.05$), indicating that the relationship between biodiversity and ecosystem functionality evolves with increasing soil age. It provides a valuable starting point for further research aimed at unraveling the intricacies of these relationships and improving our understanding of ecosystem dynamics over time.

Ln 102: In the study described in this manuscript, only undisturbed ecosystems were studied? If not, does this sentence make sense?

18. In this study, all locations were relatively undisturbed (with the exception of Taiwan). To mitigate potential influences of distinct vegetation types (cropland, forest, grassland, and shrubland) on within-site soil BEF relationships, a mixed-effect model was employed, incorporating vegetation type as a random factor. The results showed a consistent declining relationship between

within-site soil BEF relationships and soil age, substantiating the overall coherence of the statement in this sentence.

We understand the reviewer's concern regarding significant disturbances that induced vegetation destruction, such as the conversion of forest to cropland, which may potentially reset ecosystems to younger states. These disturbances may confound the effects of soil age on soil BEF relationships. By selecting ecosystems of varying ages while controlling for other factors like vegetation and parent material, the study aims to isolate the effect of soil age on soil BEF relationships. This careful selection ensures that the observed within-site changes in soil BEF relationships are attributable primarily to changes in soil age, thereby enhancing the study's validity.

We acknowledge that natural ecosystems undergo disturbances like fire and grazing, may result in regression of ecosystem to a younger state and reduce soil organic matter and microbial biomass. Our results on soil development, as illustrated by chronosequence indicate that these disturbed ecosystems may rejuvenate the importance of soil biodiversity to support function. We have discussed these important points in the discussion section (lines 252-261).

Ln 126 and materials and methods – PLFA was analyzed on dried soil samples. What evidence do the authors have (and could be cited) do the authors have that soil drying before PLFA renders valid results?

19. We appreciate the reviewer's concern. In our study, we estimated the microbial biomass in soil by analyzing PLFAs extracted from freeze-dried soil samples. The choice of utilizing freeze-dried soil samples instead of air-dried ones was driven by the benefits of rapid freeze-drying, which minimizes cellular damage and preserves the structure of cells and sensitive biomolecules. This approach is particularly advantageous for measurements of biological attributes. Additionally, it is noteworthy that PLFAs serve as biomarkers that are specific to living microbial cells. Therefore, we measured PLFAs as representative as microbial biomass using freeze-dried soil samples following numerous previous studies⁴⁻⁵. We have provided detailed information on the methodology and associated citation in lines 312-313.

Ln141-146: question arises about the role of BNF in early successional stages, not mentioned so far,

20. We appreciate this insightful comment. As responded above, we fully recognize the importance of biological N fixation, especially at the early successional stage of soil development. However, obtaining quantitative estimates of biological N fixation on a large scale poses challenges primarily due to limitations in current methodologies. In this study, the primary objective was to investigate variations in soil BEF relationships as soil develops. To achieve this, we focused on a selection of ecosystem functions such as soil respiration, decomposition, nutrient cycling, and climate regulation. While incorporating these fundamental functions, we concur that broader dimensions of ecosystem functions, including biological

N fixation, need to be integrated into the multifunctionality frameworks to provide a more holistic understanding. Despite our focus on these selected functions, this study serves as a foundational stepping-stone, demonstrating the possibility of multifunctionality assessments through the integration of various key ecosystem processes, such as soil respiration, decomposition, nutrient cycling, and climate regulation, amongst others. We have carefully discussed these limitations in the revised manuscript (lines 126-134).

Ln166-167: and what about geological origin? type of parent material?

21. The type of parent material varied across different chronosequences (across 16 chronosequences), although within each site (chronosequence), the parent material remained consistent. To mitigate the potential confounding effects, we employed a mixed-effect model by introducing parent material as a random factor. The results showed that the observed declining patterns of soil BEF with increasing soil ages still held true even after accounting for the influence of the parent material. These findings suggest that the parent material does not significantly influence the general pattern of soil BEF across soil chronosequences. We have discussed the influences of parent material in lines 177-185.

Ln 176-177: The study included agricultural soils if I understood correctly, what do the results show about this relationship in agricultural soils (frequent disturbance, analogous to early successional stage, low Organic and microbial C)? In this sense, I would also encourage the authors to elaborate Ln 231-234 a bit more.

22. Yes, our study includes a sedimentary soil chronosequence wherein a permanent crop was established. We acknowledge your concern regarding disturbances (e.g., anthropogenic activities, fire, and grazing) that destroy vegetation, reduce soil organic matter, and diminish microbial biomass, potentially resetting the ecosystem to an early stage. These disturbances may have important influences on the dynamics of soil BEF relationships by reducing organic matter and bringing back the dependence of soil biodiversity to support function. We believe that future work is needed to explore the contribution of disturbance to the soil BEF. We stated that “Importantly, disturbances of ecosystems can set back the development, causing a regression to early successional changes and leading to reductions in soil organic C and microbial biomass. Consequently, disturbances reducing soil organic matter and microbial biomass may rejuvenate the importance of soil biodiversity to support function. In our study, all locations were relatively undisturbed (with the exception of Taiwan) so future work is needed to delve further into this hypothesis.” Please see our additions in lines 252-258.

Ln180: or ecosystem (after disturbance)

23. Revised as suggested (line 253).

Ln231-234: can be elaborated more

24. We have elaborated the influences of disturbances on within-site soil BEF relationships as suggested (lines 252-258).

Ln 252-254: Please explain what original data the selection was based on.

25. The detailed information on 16 soil chronosequences, including soil age, climate, vegetation, parent material and topography were showed in Supplementary Table 1.

Ln 262: only 0-10, what about invertebrates??

26. Distinct invertebrates inhabit various horizons of the soil profile. For instance, some invertebrates, including burrowing organisms like ants and beetle larvae, may be present in the subsoil. Nevertheless, the topsoil layer (0-10 cm) is commonly the most biologically active layer. Diverse invertebrate taxa such as earthworms, springtails, and mites, are frequently found in this layer due to the presence of organic matter and plant roots ⁶. Additionally, considering the large scale of the present study, we have focus only on topsoil, taking into account practical considerations, such as time, budget, and labor constrains. Further studies in subsoils are warranted to corroborate the findings in this study.

Ln 267: biochemical or physical and chemical?

27. We have changed “biochemical” into “physical and chemical” in line 295.

Ln 268: experiment of analysis?

28. We have changed “molecular experiment” into “molecular biology analysis” in line 296.

Ln 283: Change “extracted” into “extraction”.

29. Changed as suggested in line 311.

Ln 284-285: on dried soil?? How is that possible? expressed as?

30. In this study, PLFAs were extracted from freeze-dried soil samples. We have clarified this in lines 312-313. The utilization of freeze-dried soil samples minimizes cellular damage and preserves the structure of cells and sensitive biomolecules, and thus have been widely used in the PLFAs measurements ⁴⁻⁵. Additionally, we changed “measured as” into “expressed as” following your suggestion.

Ln 312-313: non fertilized systems? why not BNF?

31. In our study, all locations were relatively undisturbed, with the exception of a sedimentary soil chronosequence wherein a permanent crop was established. We have re-analyzed data using a linear mixed-effect model, by incorporating different fertilized and non-fertilized ecosystem types as a random factor. Furthermore, we agree that disturbances such as deforestation, fertilizer, may

have influences on within-site soil BEF relationships. Nevertheless, future work is needed to delve further into this hypothesis. We have added texts to elaborate these potential influences in the discussion section (lines 252-258).

Regarding the absence of BNF, as respond above, BNF was not measured due to significant challenges associated with obtaining large-scale quantitative estimates. Furthermore, our study selectively focused on a subset of key ecosystem functions, including soil respiration, decomposition, nutrient cycling, climate regulation etc.. We recognize that this selection may influence the assessment of the relationship between soil biodiversity and multifunctionality. Consequently, it is imperative to incorporate variables targeting broader aspects of ecosystem functions, notably biological nitrogen fixation, into multifunctionality frameworks to enhance the robustness of the conclusions derived from this research. Nevertheless, this study provides a case study illustrating the possibility of evaluating multifunctionality by incorporating multiple essential ecosystem functions including soil respiration, decomposition, nutrient cycling, water and climate regulation etc. We have carefully discussed these additional critical functions and limitations of this study in the revised manuscript (lines 127-134).

Ln 317: Was this measured in the lab or in the field???

32. This experiment was conducted in the lab. We have clarified this in line 345.

Ln 373-374: based on what original data? but what about origin? This was ignored?

33. The original data of soil ages was presented in the Supplementary Table 1. Furthermore, we conducted additional analyses to examine the influence of chronosequence origin on within-site soil BEF relationships. Specifically, a mixed-effect model was employed when analyze the relationship between the within-site soil BEF and age, incorporating chronosequence origin as a random factor. Moreover, we evaluated the differences of soil BEF relationships across distinct chronosequence origins. The results indicate that chronosequence origin had no significant influences on the pattern of soil BEF relationships with increasing soil age. Please see our supplementary discussion on this in lines 177-182.

Ln 387: Change “salinaty” to “salinity”

34. Changed as suggested (line 417).

Fig 1: The polynomial curve fitting doesn't seem to make any sense at least for the majority of those curves where a linear (or exponential) relationship seems to occur. Maybe only the figure on Pathogen control seems to be non-linear.

35. We have changed the polynomial curve for WHC and ARGs control to linear relationships as suggested. Please see our changes in Fig. 1.

Ln 570: Would be good to know which functions related more or less strongly to biodiversity. Could the authors provide some results on individual functions rather than composite indices alone?

36. Thanks for this suggestion. The relationships between soil biodiversity and individual functions were depicted in Fig. 1 and Fig. S8. Our results indicate that the individual functions supported by soil biodiversity, including soil respiration, WHC, and AMF diminish as soil ages, whereas the support for lignin degradation increases with increasing soil age. Moreover, the relationship between soil biodiversity and AMF exhibits a negative correlation with SOC and microbial biomass, suggesting that in the soils characterized by lower SOC and microbial biomass, AMF function exhibits a greater dependence on soil biodiversity. We have incorporated these results associated with individual functions in lines 121-124.

Ln 590-591: what relation between soil age and ecosystem age . for example agricultural soils may be set back in ecosystem succession despite having an older soil age.

37. This is an important point. We agree with the reviewer that soil age and ecosystem development may not always coincide. In this study, all locations were relatively undisturbed, with the exception of a sedimentary soil chronosequence wherein a permanent crop was established. We have re-analyzed data using a linear mixed-effect model, by incorporating different vegetation type as a random factor. Furthermore, we have discussed the potential influences of disturbances on within-site soil BEF relationships in lines 252-258.

Ln 591: But is it in line with data presented that Higher SOC stimulates Multifunctionality

38. We agree with your perspective on this point. The higher SOC may stimulate multifunctionality by providing more labile components of energy and nutrient, consequently reducing the dependency of multifunctionality on soil biodiversity. We have provided further clarification on this point in lines 149-155. Thanks again for all the constructive comments.

Reviewer #2 (Remarks to the Author):

Ref.: Ms. No. NCOMMS-23-49193

Geologically younger ecosystems are more dependent on soil biodiversity for supporting function

The study by Feng et al. examines the relationships between biodiversity and ecosystem function (BEF) in the soil across 87 ecosystems distributed globally, which encompass a wide range of soil ages, ranging from centuries to millennia. Their findings suggest that long-term soil development leads to a significant decoupling of the relationships between BEF. This implies that in younger ecosystems with less

developed soils, there is a stronger reliance on soil biodiversity for proper soil functioning. Overall, this research provides valuable insights into the relationships between BEF as soil evolves over millennia across a global scale. These insights have the potential to enhance our ability to predict how changes in soil biodiversity can affect various ecosystem functions in a dynamically changing world. The paper is generally well-organized and presents clear hypotheses. However, before its publication, the authors should clarify some areas of confusion in the results.

39. Thank you very much for these positive feedbacks on this manuscript. We have carefully addressed all the confusions involved in the description of method, clarification of results, and confusions associated with drylands to ensure a more robust and understandable presentation of our findings (e.g., lines 69, 74-75, 173-185).

General comments:

1 The introduction section lacks critical information on the methodology of the study. For instance, there is a lack of clarity regarding the number of chronosequences included in this study and the specific methodologies employed to quantify the relationship between soil biodiversity and ecosystem functions.

40. Thanks for these insightful comments. We have provided additional information on the number of chronosequences studied, along with the explanation of methodologies employed to assess soil BEF relationships. Please see our revisions in lines 69, 74-75, 400-404.

2 The results regarding drylands are intriguing but require further clarification. The authors state that the capacity of soil biodiversity to support ecosystem functions will be weaker in older productive ecosystems, such as drylands (Line 57-59). However, it is essential to acknowledge that drylands do not uniformly consist of young soils with limited soil age. Drylands can have weakly developed or 'pedogenic young' soils due to weak leakage caused by water limitation. Therefore, even after millions of years of development, soils in drylands may remain weakly developed, potentially contributing to the non-significant reduction of soil BEF relationships across chronosequences in drylands. This explanation should be further elucidated throughout the manuscript.

41. We acknowledge the need for further clarification regarding the dynamics of within-site soil BEF relationships as soil development in drylands. We have clarified that water limitation can lead to insufficient soil development due to restricted water availability for soil weathering. This may contribute to the absence of decline in within-site soil BEF relationships along chronosequences. All clarifications related to drylands have been carefully addressed throughout the manuscript. Please see our revisions in lines 173-185.

3 The manuscript omits important definitions, such as 'multifunctionality', which should be provided at the first instance of their presentation in the text. It is

imperative that these key terms are comprehensively defined and consistently used throughout the manuscript for the sake of clarity and precision.

42. We have defined these important terms as suggested throughout the manuscript (e.g., lines 40, 58, 86).

Minor comments:

Line 40: Please define BEF for the first time of presence.

43. Defined as suggested (line 40).

Line 50-51: Please define 'net soil functions'.

44. We have changed "net soil functions" into "essential soil functions" in the revised manuscript (line 51).

Line 52: As mentioned above, the term 'BEF relationship' should be defined for the first time of presence.

45. Revised as suggested (line 52).

Line 59: While it is true that drylands are typically less productive, it is important to note that not all dryland soils are inherently young in terms of soil age. Drylands may exhibit 'young' pedogenic characteristics, primarily attributed to weak soil development as a result of poor leaching and water scarcity.

46. We agreed with you that drylands may be pedogenically "young" due to water limitation. We have removed the inappropriate statement in line 59. Additionally, we have integrated the related information into the discussion section to ensure a more accurate representation of soil BEF relationships in drylands (lines 173-185).

Line 61: Please define 'multifunctionality' for the first time of presence.

47. Defined as suggested (line 64).

Line 65: Unclear. How many chronosequences do you have?

48. There were 16 soil chronosequences. We have provided this detailed information in line 69.

Line 71: Unclear. How the authors quantify soil BEF? Was Spearman's correlation coefficient used? Please clarify.

49. Yes. Spearman's correlation coefficient was employed to quantify soil BEF relationships. We have included the corresponding details in line 74-75.

Line 86: This paragraph encapsulates the main findings of the manuscript. Therefore, it may be advisable to relocate the subtitle to the second paragraph.

50. Revised as suggested (line 89).

Line 87: While drier ecosystems may have soils that are old in terms of their formation, they can also be geologically young due to weak soil development. It is necessary to provide further clarification on this matter.

51. This is an important point. We have clarified that “In drylands, soils may be weakly developed even when aged due to water restrictions on soil weathering, soil organic C accumulation and biomass production” (lines 173-176).

Line 139: Please supplement the relevant references.

52. Added as suggested (line 155).

Line 165-167: This is an interesting point, but it requires further elucidation. The relatively constant soil BEF relationships along chronosequences in drylands may be attributed to the limited soil development in drylands.

53. Thanks for this constructive comment. We have elucidated that “In drylands, soils may be weakly developed even when aged due to water restrictions on soil weathering, soil organic C accumulation and biomass production⁷⁻⁸. Therefore, the negative influence of soil age on BEF may be limited in these ecosystems with reduced soil development.” (lines 173-176).

Line 219: Other factors, such as soil pH, also varied significantly across chronosequences. Importantly, soil acidification as soil ages may also induce changes in soil microbial biomass and diversity. How will changes in soil pH influence soil BEF relationships?

54. Thanks for this insightful comments. We have discussed the potential influences of soil pH and other factors on soil BEF in the revised manuscript (242-244, 258-261).

Line 272: Please define the abbreviate for the first time of presence.

55. Revised as suggested.

Line 369-370: The associated results should be provided in the Results section.

56. Provided as suggested (lines 98-100).

Line 376-377: Which models do you selected? Please clarify.

57. Clarified as suggested (lines 400-402, 406-408).

Line 394: Again, ‘SEM’ should be defined for the first time of presence.

58. Defined as suggested (line 86).

Reviewer #3 (Remarks to the Author):

The manuscript by Feng et al. assesses the effects of soil biodiversity on ecosystem functioning (water retention, decomposition, relative abundance of pathogens and mutualists) across 87 sites around the globe that vary in soil age (soil pedogenesis stages) which also co-vary with vegetation cover and biomass. The data show that geographically younger soils exhibit stronger positive soil biodiversity-ecosystem functioning relationships than older soils. The authors argue that this is largely because of the lower plant cover, soil organic C accumulation and microbial biomass production in younger dryland soils making them more susceptible to reduced functioning with soil biodiversity loss. Conversely, in older soils there is more biological activity and inputs via plants that may relatively diminish the importance of soil biodiversity for ecosystem function. The study is very thorough in its data collection analyses and is generally well written. The study reveals important insights into the ecological function of soils, its biodiversity and community composition as ecosystems develop along a chronosequence and in particular I think it can provide novel insights into the role of soil biodiversity and community composition with relation to fast versus slow ecosystem processes, and how ecological complexity increases with soil age. While I agree with statements emphasizing the need for soil conservation in drylands, there are a few points that need some clarification / consideration with regards to the diversity & redundancy vs. compositional role in 'older' soils.

59. Thank you for your positive comments on our manuscript. We appreciate your recognition of the thoroughness in data collection and analysis, as well as the novel insights our study provides. We acknowledge your point regarding the need for clarification of the role of biodiversity and redundancy vs. compositional role in 'older' soils. We have addressed this concern by providing additional explanations and discussions in the revised manuscript. Please see our additions in lines 114-120, 230-234.

Main points

1. How comparable are the diversity gradients between younger and older soils? I assume older soils are more diverse? If biodiversity-function relationships are asymptotic then are the younger soils at the lower steeper end of the curve and older at the higher less steep end of the curve?

60. Yes. The older soils are generally more diverse than younger soils, as shown by the linear increases of multi-trophic biodiversity (including invertebrates, protists, fungi, and bacteria) with increasing soil age. We have incorporated the corresponding results into Fig. S2. Additionally, we added text to elucidate the associated results in lines 115-117. Specifically, we state that "Consequently, in younger soils with lower soil biodiversity (Fig. S2), the increase in species diversity contributes to more efficient resource utilization and facilitates the enhancement of multiple ecosystem functions."

2. On the issue of functional redundancy: This is an interesting point (L112-114), but I would be very cautious about how "redundancy" is used here for fear of misinterpretation because redundancy fades with the consideration of temporal

and spatial patterns in functioning and mode functions considered. Because there is no positive BEF relationship in some soils based on the specific functions measured does that mean that they are truly functionally redundant in the ecosystem, or that they the diversity and additional compositional components are not related to the functions you measured? The concern here being that this may be interpreted to the general public as that soil biodiversity is not important in well established soils, thus we shouldn't be concerned about soil biodiversity in these systems. I don't think this is the message you are trying to convey. Theoretically, if there was a lot of true functional redundancy in that the organisms perform all the same functions in the same space at the same time would not resource competition lead to their divergent functional evolution? I think these statements in the text need to be changed for the relevance to the functions measured here with consideration for what "redundancy" means.

61. Thanks for these constructive comments. We acknowledge the potential misinterpretation of "functional redundancy", especially when considering temporal and spatial patterns in functions, mode functions considered, and the selection of specific functions that measured. In the revised manuscript, we have removed the potential misinterpretations regarding "functional redundancy". Instead, we stated that "Consequently, in younger soils with lower soil biodiversity (Fig. S2), the increase in species diversity contributes to more efficient resource utilization and facilitates the enhancement of multiple ecosystem functions. In well-established older soils, however, the contribution of soil biodiversity to support function may be less noticeable given the legacy of millions of years of organic matter and microbial biomass accumulation which can now feed the ecosystem with resources." (lines 115-120).

We acknowledge that the choice of selected functions may influence the evaluation of the relationship between soil biodiversity and multifunctionality. Therefore, we highlight that integrating variables associated with broader dimensions of ecosystem functions, such as biological N fixation into the multifunctionality frameworks in the future will strengthen the conclusions drawn here. We have carefully discussed this limitation in lines 127-134.

3. Alternative reasons for the lack of significantly positive BEF relationships in 'older' soils: L119-120: Ok. Or is it that in younger systems they are key for these functions, but in older ecosystems may play a greater diversity of functions not measured here and thus leading to a lack of strong BEF relationships based on the functions measured here. These sweeping statements can be misleading, when not all facets have been carefully considered. Then I see later on this is touched on L180-on that its not always biodiversity, but composition. It would be good to make this clear that diversity and composition may have varying roles as the soil ecosystem evolves where in resource poor biodiversity, having more, is better, while in older more complex ecosystems where there diversity is high composition is likely more important for given functions than just having more (i.e. the insurance hypothesis).

62. This is an important point. We have carefully discussed the influences of the choice of different functions on the relationship between soil biodiversity and

multifunctionality. Specifically, we have stated that “We acknowledge that, in our study, the choice of functions may influence the evaluation of soil BEF relationships. Therefore, we emphasize the necessity of incorporating variables associated with broader dimensions of ecosystem functions, such as biological N fixation, food production or polittization (among many others), into the multifunctionality frameworks to reinforce the robustness of conclusions in this study.” (lines 126-130). Additionally, we have highlighted the pivotal role of microbial community composition in driving given ecosystem functions following your suggestions. We stated that “Thus, a high level of overall biodiversity may not necessarily contribute equally to all measured functions in older and more complex ecosystems. Instead, certain functions may become more dependent on specific compositions within the community. These results underscore the pivotal role of soil community composition in driving fundamental ecosystem functions in older soils.”. Please see our changes in lines 230-234.

Minor points

L26-27: this sentence is awkward / incomplete. Reword.

63. Revised as “We further show that increases in plant cover, soil carbon and microbial biomass as ecosystems develop, particularly in wetter conditions, lessen the critical need of soil biodiversity to sustain function.” (lines 26-28).

L41: I am not sure “tools” is really the appropriate term. Tools infers for use / manipulation. Why not “functions” or “biogeochemical processes”

64. Changed “biogeochemical tools” to “functions” as suggested (line 41).

L86: seems a bit abrupt to start with stating what the study shows without first stating the result. It would be better to start with L90 stating what the data show, then interpret for discussion point (eg. “our study could show that plant cover, soil C ...”).

65. Thanks for this comment. We have now revised the sentences to start with the main results of the study. “Our work provides new evidence that soil biodiversity is more important for supporting function in younger and less productive ecosystems” (lines 89-90).

L91: Not clear to me what “increase in weather environments” means.. reword.

66. Sorry for this unclear expression. The phrase “increase in weather environments” indicate an increase in soil weathering. However, we have removed the associated sentences to focus more on the main results of this study here (lines 90-94).

L94-97: again this statement is meaningless here when there has been no evidence provided first to support these statements overselling the study without first presenting the facts.

67. Thanks for this comment. We have removed these inappropriate presentations in the revised manuscript.

L98: This is the important stuff to state first. The above paragraph adds nothing meaningful to the text and can be removed.

68. Thanks for this valuable comment. We have now started with the main findings of the study following your suggestion. Moreover, we have carefully removed any statements potentially overselling the results of this study. As a result, this paragraph now contains only a brief summary, guiding the key findings of our study. This concise statement provides a comprehensive overview of our research without exaggeration of unnecessary embellishments (lines 89-96).

L105: This is inevitable based on the threshold method where the lowest threshold supports all functions across the diversity gradient and non are supported at all levels of diversity with the maximum threshold. What is interesting is the range in the thresholds and where it is at its maximum: T_{max} , T_{min} , and T_{mode} . Here it seems to be steepest at 50 and 75% indicating that greater biodiversity supports fewer functions above average as the soil ecosystem ages if I am interpreting this figure properly.

69. We appreciate the insightful interpretation you provided on the figure. As you pointed out, the observed pattern, with the steepest range in thresholds at 50% and 75%, suggests that as soils age, greater biodiversity tends to support fewer functions above the average threshold. This aligns with the idea that in older and more complex ecosystems, a high level of biodiversity may not necessarily contribute equally to all measured functions. Instead, certain functions may become more dependent on specific compositions within the community rather than overall biodiversity. We have incorporated associated interpretations in the discussion section in lines 108-110, 230-234.

Fig. 1B. I am not sure you need a legend/specific colours for soil age when it is the x-axis. It could removed to declutter the panel.

70. We have removed the legend and colors as suggested. Please see our changes in Fig. 1B.

L256 - Is soil age related to climate (Sorry if I missed it)? What if in Fig 2 climate predicted soil age (or vice versa or allowed to covary)? If climate is included as a co-variate (or data de-trended for climate) in the analyses change results?

71. We appreciate this comment. In our study, soil age and climate are treated as independent variables. Soil age, specifically the time since initial soil formation, is distinct from climate, although we acknowledge that climate is a significant factor influencing soil development. Our structural equation modeling (Fig. 2) have been carefully designed to account for the potential influences of both soil climate and soil age on within-site soil BEF relationships. Therefore, the results presented in our study have considered both two factors comprehensively,

ensuring that the confounding effect of climate has been simultaneously incorporated.

L307: Not sure if these are all truly functions (i.e. enzyme activity) or ecosystem attributes (i.e. water holding capacity) that reflect its functioning. It's a picky thing that seems to come up these days, but simply stating that these measures 'reflect biological functioning' would help with clarity.

72. Thanks for this insightful comment. We have toned down the language to state that these measures represent proxies reflecting the biological functioning of the ecosystem (lines 336-337). Thanks again for all your constructive comments!

References:

1. Ladha JK, *et al.* Biological nitrogen fixation and prospects for ecological intensification in cereal-based cropping systems. *Field Crop Res* **283**, 108541 (2022).
2. Soper FM, Simon C & Jaus V. Measuring nitrogen fixation by the acetylene reduction assay (ARA): is 3 the magic ratio? *Biogeochemistry* **152**, 345-351 (2021).
3. Odum EP. The Strategy of Ecosystem Development. *Science* **164**, 262-270 (1969).
4. Lewe N, *et al.* Phospholipid fatty acid (PLFA) analysis as a tool to estimate absolute abundances from compositional 16S rRNA bacterial metabarcoding data. *J Microbiol Meth* **188**, 106271 (2021).
5. Wang S, *et al.* Co-analysis of cucumber rhizosphere metabolites and microbial PLFAs under excessive fertilization in solar greenhouse. *Front Microbiol* **13**, (2022).
6. Bardgett RD & van der Putten WH. Belowground biodiversity and ecosystem functioning. *Nature* **515**, 505-511 (2014).
7. Feng J, *et al.* Phosphorus transformations along a large-scale climosequence in arid and semi-arid grasslands of northern China. *Global Biogeochem Cy* **30**, 1264-1275 (2016).
8. Wang L, *et al.* Dryland productivity under a changing climate. *Nat Clim Change* **12**, 981-994 (2022).

REVIEWERS' COMMENTS

Reviewer #1 (Remarks to the Author):

I believe the authors have done a very good job explaining how the death with the comments of the reviewers and I recommend to accept the revised version for publication after the following minor changes.

Ln 19 (Abstract): "Soil biodiversity contains the metabolic toolbox supporting organic matter decomposition and nutrient access in the soil"=> I suggest to change "nutrient access" to "nutrient cycling".

Ln 57: I suggest to change "fundamental contribution" to "fundamental role".

Ln 26-28: In this explanation could increased functional redundancy in more diverse, older systems also play a role? See also Ln 92-93: ".. probably as a consequence of the organic 93 matter and microbial biomass reservoir built over millions of years of soil development" => What about increased biodiversity in older soil leading to increased functional redundancy?

Ln 262: What is "multidiversity"?

Reviewer #2 (Remarks to the Author):

The author made detailed revisions to the manuscript, and addressed all my concerns. I recommend accepting as it.

Reviewer #3 (Remarks to the Author):

The authors have addressed all my comments and concerns sufficiently. I do not have any further concerns with the manuscript.

Note: L130 - "politization" ? I am not sure what this refers to in this context. Typo?

Response letter

Responses to reviewer's comments:

Reviewer #1 (Remarks to the Author):

I believe the authors have done a very good job explaining how the dealt with the comments of the reviewers and I recommend to accept the revised version for publication after the following minor changes.

1. We're pleased that the reviewer is satisfied with the revisions made. We greatly appreciate the positive feedback and the diligent efforts in reviewing our manuscript. We have addressed all minor changes. Please see detailed responses as follows:

Ln 19 (Abstract): "Soil biodiversity contains the metabolic toolbox supporting organic matter decomposition and nutrient access in the soil"=> I suggest to change "nutrient access" to "nutrient cycling".

2. Revised as suggested (Line 22).

Ln 57: I suggest to change "fundamental contribution" to "fundamental role".

3. Revised as suggested (Line 57).

Ln 26-28: In this explanation could increased functional redundancy in more diverse, older systems also play a role? See also Ln 92-93: ".. probably as a consequence of the organic matter and microbial biomass reservoir built over millions of years of soil development" => What about increased biodiversity in older soil leading to increased functional redundancy?

4. Thanks for these detailed comments. We have incorporated texts to clarify the potential influences of changes in soil biodiversity on within-site soil BEF relationships. We clarified that "Additionally, the higher soil biodiversity following organic matter accumulation may lead to functional redundancy of essential functions measured in this study, further contributing to the diminished soil BEF relationships in older soils.". Please see our changes in lines 124-127.

Ln 262: What is "multidiversity"?

5. Multidiversity represents the averaged richness of invertebrates, protists, fungi and bacteria. Please see our definition in line 105.

Reviewer #2 (Remarks to the Author):

The author made detailed revisions to the manuscript, and addressed all my concerns. I recommend accepting as it.

6. Thanks a lot for the positive feedbacks and all the constructive comments to improve our manuscript.

Reviewer #3 (Remarks to the Author):

The authors have addressed all my comments and concerns sufficiently. I do not have any further concerns with the manuscript.

7. We're glad that the reviewer is satisfied with our revisions. Thanks a lot for these positive comments.

Note: L130 - "politization" ? I am not sure what this refers to in this context. Typo?

8. We are referring to "policy" in this context. We have revised this in line 137. We are grateful to the reviewer for this detailed comment.